# Beyond Diameter: Enhancing Abdominal Aortic Aneurysm Surveillance with Volumetric Assessments after Endovascular Aneurysm Repair (EVAR)

**DOI:** 10.3390/jcm12216733

**Published:** 2023-10-25

**Authors:** Michał Kargul, Patryk Skórka, Piotr Gutowski, Arkadiusz Kazimierczak, Ireneusz Wiernicki, Paweł Rynio

**Affiliations:** Department of Vascular Surgery, Pomeranian Medical University in Szczecin, Al. Powstańców Wielkopolskich 72, 70-111 Szczecin, Polandireneusz.wiernicki@pum.edu.pl (I.W.)

**Keywords:** sac enlargement, growth of sac volume, endoleaks, aorta, abdominal aortic aneurysm, volume measurement, remodeling of AAA, surveillance after EVAR, negative remodeling, EVAR

## Abstract

This study aimed to investigate the relationship between maximum transverse diameter (MTD) and volume measurements in patients who underwent reoperations after endovascular aneurysm repair (EVAR), and their association with the occurrence of endoleaks. The study included 51 patients who underwent EVAR and subsequent re-operations caused by endoleaks type I–III. In some number of events, multiple re-operations were needed. MTD was measured using the Horos software, and segmentations of the AAA were performed using 3D Slicer. This study first evaluated post-operative computed tomography angiography (CTA) to measure MTD and volume. Then, similar measurements were made in the control scan for re-operation qualification. Negative remodeling (increase in MTD and/or volume) was observed in 40 cases using MTD, and 48 cases using volume measurements. The volume measurement showed lower missed negatives than MTD, indicating its effectiveness in screening for negative remodeling (*p* < 0.001). Combining both methods identified 51 negative remodeling cases and 8 positive changes, with a higher sensitivity compared to MTD alone. The volume of the sac did not predict specific endoleak types. Decreases in MTD were observed in smaller sacs, with smaller volume changes. Volume measurement is a valuable screening tool, and combining MTD and volume enhances sensitivity. However, sac volume does not predict endoleak type.

## 1. Introduction

An abdominal aortic aneurysm (AAA) is a life-threatening dilatation of the aorta in the abdomen that can be repaired with either an open surgical repair (OSR) or an endovascular repair (EVAR). As of right now, intervention thresholds are based on maximum transverse diameters (MTD), where surgical treatment should be considered if the MTD reaches 5.5 cm for males, and 5.0 cm for females [1]. EVAR is a leading treatment method in patients with diagnosed AAA [2]. In recent studies, the long-term mortality and rupture risk of EVAR are similar to those of OSR [3].

Endoleaks in EVAR refer to blood escaping outside the graft, causing persistent flow within the aneurysm sac. The most common type is endoleak type II, accounting for 20–40% of cases [1]. It is caused by retrograde blood flow from collateral vessels, like lumbar or inferior mesenteric arteries, and often resolves on its own. Despite a maintained seal between the endograft and aorta, retrograde flow leads to sac expansion.

The primary aim of post-EVAR imaging is to detect endoleaks and stent-graft issues. Computed tomography angiography (CTA) is the standard within the first 30 days, allowing for the detection of sac enlargement, endoleaks, sealing zone integrity, stent-graft migration, limb issues, and infections [1]. However, CTA has drawbacks, including radiation and nephrotoxic contrast in patients with kidney issues. Contrast-ineligible patients can consider non-contrast CT for MTD and sac volume assessments to detect possible endoleaks [4].

The European Society for Vascular Surgery suggests a follow-up strategy after EVAR. Within 30 days, the first examination aims to detect endoleaks and confirm aneurysm exclusion. Based on this, patients are categorized into three groups: Low risk (no endoleaks, adequate stent-graft overlap)—with follow-up every 5 years. Intermediate risk (adequate overlap, but with endoleak type II)—requiring yearly imaging, preferably duplex ultrasound. High risk (inadequate overlap, and endoleak type I or III)—prompting a consideration of re-intervention based on findings [1].

Few indicators are available for assessing re-operation risk, like endoleak evidence, sac expansion, graft issues, infections, and aneurysm neck shape. MTD, relying on a single measurement from CTA scans, overlooks information about remodeling and morphology changes at the proximal and distal ends of the aneurysm. In recent studies, volume has been shown as a reproducible measurement, which can show those changes [5,6].

This research evaluated the relationship between MTD and volume between first CTA after primary EVAR and CTA scan after re-operation, volume growth rates in reference to the first CTA after primary EVAR, and the occurrence of endoleaks type I to III, based on a retrospective study of CTA data from patients in which those endoleaks appeared.

## 2. Materials and Methods

### 2.1. Study Design

This study employed a single-center, retrospective, observational design to investigate events in patients who underwent re-intervention after EVAR for endoleak type I–III for infrarenal AAA between 1 January 2010 and 31 December 2021. The study specifically focused on patients who underwent initial EVAR and subsequent elective re-operations specifically due to endoleaks type I–III. All patients received treatment using the endograft Endurant 2S from Medtronic (Santa Rosa, CA, USA), and no ancillary procedures, such as renal stenting and endoanchors, were performed during the interventions.

The inclusion criteria for this study comprised patients who underwent EVAR and subsequent re-operations at our clinic. Patients who underwent branched EVAR, fenestrated EVAR, and chimney EVAR procedures were excluded, as were those with landing to the external iliac artery. CTA scans obtained from the patients’ medical records within the hospital were considered to be eligible if they had a maximum thickness of 1 mm and included all three phases: native, arterial, and delayed. The CTA scans used for analysis were from the first CTA scan taken within 30 days after the primary EVAR, the control scan for re-operation qualification, and any additional control scan before the subsequent re-operations. Patients from nearby hospitals were referred to our clinic with CTA scans, but a thorough follow-up and control CTA were not available for all patients, as routine checks were carried out in another unit. However, patients from these external hospitals were included in the study if they met the specified criteria.

Due to the retrospective nature of the study, the need for informed consent was waived. A total of 110 computed tomography angiography scans were examined to conduct the analysis and draw conclusions.

### 2.2. Measurements

The measurement of the MTD of the aneurysm was conducted using the Horos 3.0 software (Horos Project, Annapolis, MD, USA). This was achieved through the double oblique method, which involved selecting the largest section of the AAA in a plane perpendicular to the centreline. This approach ensured reproducible measurements across all the CTA scans.

The segmentations of the AAA were performed using the 3D Slicer program [7], with particular attention given to ensuring consistency by having the same individual work on each patient’s segmentation to ensure that interobserver variability was reduced. All the segmentations were then evaluated by a vascular surgeon and the person that made those segmentations. The segmentation team consisted of three members: one physician and two students from who worked in the vascular surgery department and in-hospital 3D printing laboratory. Prior training on Horos and 3D Slicer was provided to the team, who had previous experience in performing AAA segmentations.

The volume of the AAA was obtained through the segmentation of the aneurysm sac. The process involved a semi-automated approach utilizing the tools available in 3D Slicer. The segmentation encompassed the intraluminal thrombus, calcifications, and the lumen of the aneurysm. In the case of follow-up data, additional structures such as the stent-graft and any present endoleaks were also segmented. The segmentation start point was when the infrarenal aorta diameter was ≥50% greater than the suprarenal diameter. The segmentation endpoint was typically at the level of the common iliac artery or the aortic bifurcation, depending on the extent of the aneurysm.

The comparison of the volume and MTD was based on the first CTA scan taken within 30 days after the primary EVAR, the control scan for re-operation qualification, and any additional control scan before the subsequent re-operations. A comparison was made between the following CTA scans. A graphical flowchart of study design is presented in Figure 1.

The combined method utilized both MTD and volume measurements to determine the growth of the AAA sac in either parameter. A negative remodeling of the AAA was defined as an increase in the MTD and/or volume, indicating progressive changes in the aneurysm over time.

### 2.3. Statistical Analysis

Statistical calculations were performed using the Statistica 13.3 software. Clinical and demographic characteristics were presented as frequencies and percentages of the total. Nominal variables were assessed with Fisher’s test or Chi-square test. The normality of the quantitative variables was checked with a Shapiro–Wilk test and then they were compared with a *t*-Student test or Mann–Whitney U test based on the distribution of the variables. The quantitative variables for 3 groups were analyzed with a Kruskal–Wallis test. The correlations between the diameter and volume and their changes were examined with Spearman’s rank correlation or Pearson’s correlation based on the distribution of the variables. These correlations are displayed on graphs. The significance level was set at *p* < 0.05.

## 3. Results

### 3.1. Patients’ Cohort

The study sample comprised 51 patients who met the inclusion criteria. Among them, 43 individuals (84.31%) were male, while 8 individuals (15.69%) were female. The mean age of the participants was 79 ± 7.89 years, with a range from 63 to 93 years.

The complete distribution of co-morbidities related to AAA within the patient population can be found in Table 1.

### 3.2. Differences in Native AAA Structure

The volume of AAA exhibited a strong positive correlation with the MTD in the study population (N = 59; R = 0.843625; *p* < 0.00). A graphical representation is presented in Figure 2.

A strong positive correlation was observed between the change in volume and the change in the diameter of the AAA (*p* < 0.00). A graphical representation is presented in Figure 3.

An average depiction of AAA and its changes are presented in Table 2. 

Changes in the sac of AAA according to an increase in or regression of MTD are presented in Table 3.

### 3.3. Comparison between Using MTD and Volume

Negative remodeling of the AAA sac was observed in 40 cases using the MTD method. However, when volume measurements were employed to determine the AAA growth, a higher number of cases, specifically 48, were identified as exhibiting negative remodeling. The percentage of identified cases was notably higher (*p*-value < 0.00) using the volume-based method (87.36%) compared to the MTD-based method (67.80%).

Furthermore, the volume measurement approach demonstrated a lower number of missed negative cases (decrease in MTD but growth in volume) cases of remodeling (*n* = 3; 7.50%) compared to the MTD group (*n* = 11; 57.89%). This difference in performance between the two methods was statistically significant, with a *p*-value of 0.00.

### 3.4. Comparison between Individual Parameters and Combined Method

The integration of volume and diameter measurements in the determination of the negative remodeling in AAA yielded noteworthy results. Through the combined method, we identified 51 cases (*n* = 59, 86.44%) exhibiting negative changes and 8 cases (*n* = 59, 13.56%) displaying positive changes. 

In contrast, when employing diameter measurements alone, 40 cases (*n* = 59, 67.80%) demonstrated negative changes, while 19 cases (*n* = 59, 32.20%) showed improvement (*p* < 0.00).

Likewise, the comparison between the combined method and volume measurements alone revealed that volume increased in 48 cases (*n* = 59, 81.36%), while positive changes were observed in 11 cases (*n* = 59, 18.64%) (*p* < 0.00). 

A graphical representation (Figure 4) elucidated the relative percentage of the total negative remodeling of each method.

### 3.5. Changes of MTD Corresponded to Types of Endoleak

Among the cases demonstrating regression (*n* = 19), the majority of endoleaks observed were classified as type I (63.16%, *n* = 12). Conversely, a higher incidence of endoleaks was observed in cases with an increased MTD (*n* = 40). Notably, endoleaks type II were the most prevalent (52.50%, *n* = 21), followed by endoleaks type I (35%, *n* = 14), indicating a statistically significant association (*p* = 0.02). The mean changes in diameter, categorized by the type of endoleak, are provided in Table 4.

### 3.6. Changes of Volume Corresponded to Types of Endoleak

A substantial proportion of endoleaks (81.36%, *n* = 48) were observed in cases where the volume of the AAA increased. Within these cases, endoleaks type II were observed in 39.56% (*n* = 19) and type I in 41.67% (*n* = 20) of instances. However, this relationship did not reach statistical significance (*p* = 0.13). Notably, the mean changes in volume, stratified by the type of endoleak, can be found in Table 4.

## 4. Discussion

### 4.1. Role of This Study

This study found that combining volume and diameter measurements is more effective for detecting AAA sac changes in patients with endoleaks types I–III compared to using volume or diameter alone. This highlights the need to integrate both measures in clinical practice.

These findings are crucial, advocating for the use of both volume and diameter assessments in AAA monitoring after EVAR. This approach enhances understanding of disease progression, aids decision making, and anticipates complications using 3D segmentation techniques.

The study also revealed a strong positive correlation between AAA volume and MTD, showing that initial sac size can impact the response to treatment. However, no significant link was found between volume and endoleaks, indicating that other factors influence reinterventions and positive volume changes do not guarantee endoleak absence.

### 4.2. Volume vs. Diameter

When comparing volume measurement to diameter measurement, our study underscores the superior sensitivity of volume assessments post EVAR. Wever et al. showed that MTD is less sensitive than volume measurements after EVAR in more than one third of cases, and diameter measurements alone fail to identify size changes [8]. However, our results revealed that combining both volume and diameter measurements further enhances sensitivity, accurately identifying cases necessitating reoperation at a rate of 86.44%, as opposed to 81.36% when using volume measurements alone.

Our research aligns with a systematic review focusing on a comparison between diameter and volume measurements in assessing AAA sacs [9]. Notably, the majority of reviewed studies (17 out of 19) concurred that volume measurements offer a superior parameter for characterizing AAA sac morphology. However, it is worth noting that, despite this consensus, clinical decisions surrounding aneurysm repair predominantly rely on diameter measurements, which potentially limits the primary use of volume measurements [9].

Van Keulen et al. indicated that using only MTD as cut off parameter would miss 50% or more of cases with increased volume [6]. This study did not show those results, as MTD caught 83.3% of cases of increased volume, although we agree that volume is a better parameter for showing the negative remodeling of AAA than MTD. However, by using both methods, we can improve even further the sensitivity of screening. There is still need for more a homogeneous methodology and standardized protocol of measurements.

### 4.3. Role of Volume in Surveillance

The recognition of volume measurements as a potentially superior parameter for accurate assessment in EVAR dates to 1996 [10]. Despite this early recognition, the field still relies on MTD as the surveillance standard. The lack of consensus on defining volume changes, their comparability to MTD, and their impact on post-operative complications adds complexity.

Researchers have made efforts to address these questions. Lee et al. and Bastos et al. proposed that a volume reduction of ≥10% six months post-EVAR could serve as a predictor of success [11,12]. Bargellini et al. indicated that the absence of a volume decrease in the sac, exceeding 0.3%, is highly indicative of endoleaks [13]. Other studies have shown that a volume increase of ≥12% corresponds to a ≥5 mm increase in aortic diameter [14].

Our research contributes to this body of knowledge by demonstrating that combining MTD and sac volume provides a more sensitive screening tool for identifying patients in need of reoperation and detecting negative sac remodeling. This approach outperforms the use of MTD alone. Notably, Montelione et al. reached similar conclusions in their own research, noting that a larger preoperative MTD and volume, as well as negative early sac remodeling in AAA, are associated with worse long-term outcomes in EVAR [15]. Incorporating volume measurements in assessing infrarenal AAA patients greatly enhances outcomes by enabling early issue detection, tailored follow-up, and advancing research. However, standardized guidelines for implementation are needed. Future research should explore volume changes, diameter, and post-operative complications for better criteria in EVAR patient care.

### 4.4. Limitations of Study

This study has various limitations: its retrospective design may lead to missing patient data, and the small sample size (51 patients) could impact its statistical power. Being a single-center study limits its generalizability, as different centers have varying approaches and patient populations. The exclusion of certain EVAR procedures and patient subgroups further limits representation. The study’s focus on specific time points and the lack of longitudinal follow-up may hinder observing long-term changes.

Another limitation of this study is the exclusion of a group of patients who did not undergo reintervention after EVAR. During the study period, there were 71 reinterventions observed, with 59 being included in our analysis. However, 12 reinterventions had to be excluded due to the unavailability of CTA data. These exclusions resulted from initial post-surgery CTA scans being conducted outside our vascular center and thus being inaccessible for analysis. We acknowledge the potential for selection bias associated with this exclusion. It is essential to note that the study’s findings specifically apply to the group that underwent reintervention, and the exclusion of the second group may limit the generalizability of the results.

Creating a control group without reintervention is challenging, as matching these patients without affecting the study’s integrity is complex. This is primarily due to patients with a stable MTD but increased sac volume, especially in the presence of endoleaks type II or type V, who may have been misclassified in the observation group. This potential misclassification could have falsely inflated the rate of sac volume increase in the control group.

Lastly, while established software tools were utilized for measurements, inherent limitations in the accuracy of MTD and volume measurements in CTA scans may affect the precision of the findings.

A prospective study with a diverse, larger patient group is essential. Volumetric assessment should be a key EVAR eligibility criterion for a better accuracy, capturing aneurysms’ multidimensional nature. A prospective design, larger population, and diverse patients support robust, personalized medicine. Collaboration and rigorous data collection are essential for high-quality, reliable results.

## 5. Conclusions

The combined method of using both MTD and sac volume is a more sensitive screening approach compared to using MTD alone. However, sac volume alone does not provide information on the specific type of endoleak that may occur or be present. It is worth noting that, on average, decreases in MTD were observed in smaller sacs, and the changes in volume were on a smaller scale.

## Figures and Tables

**Figure 1 jcm-12-06733-f001:**
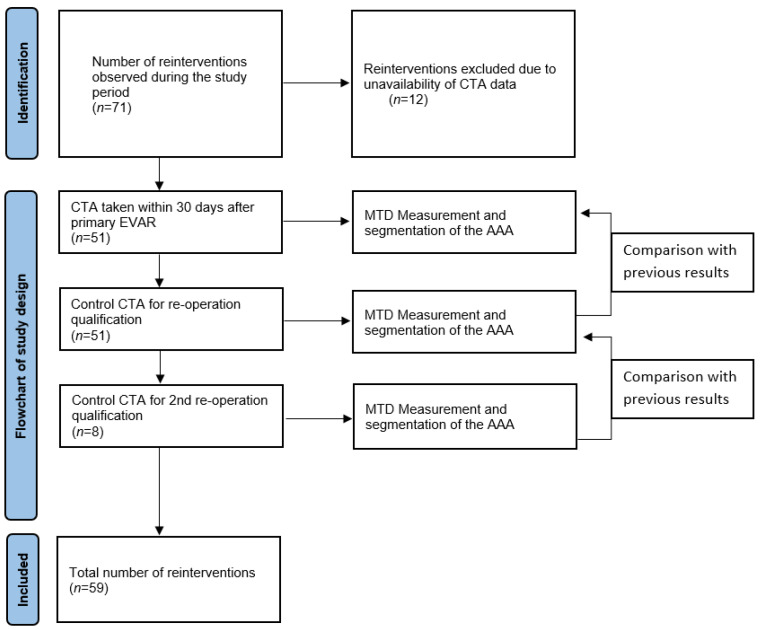
Flowchart.

**Figure 2 jcm-12-06733-f002:**
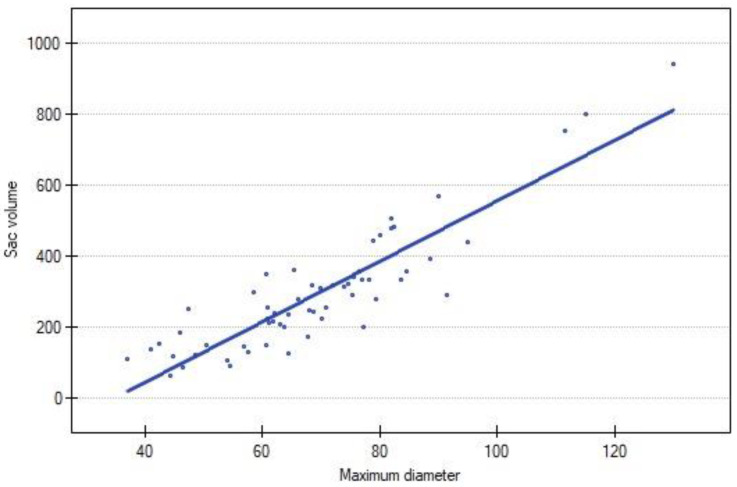
Correlation between sac volume and maximum diameter of AAA. Blue dot represents each reintervention.

**Figure 3 jcm-12-06733-f003:**
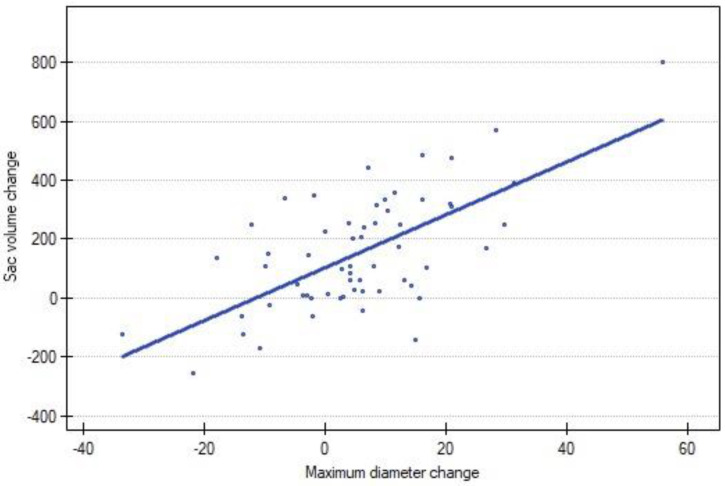
Correlation between sac volume change and MTD change between following CTA scans. Blue dot represents each reintervention.

**Figure 4 jcm-12-06733-f004:**
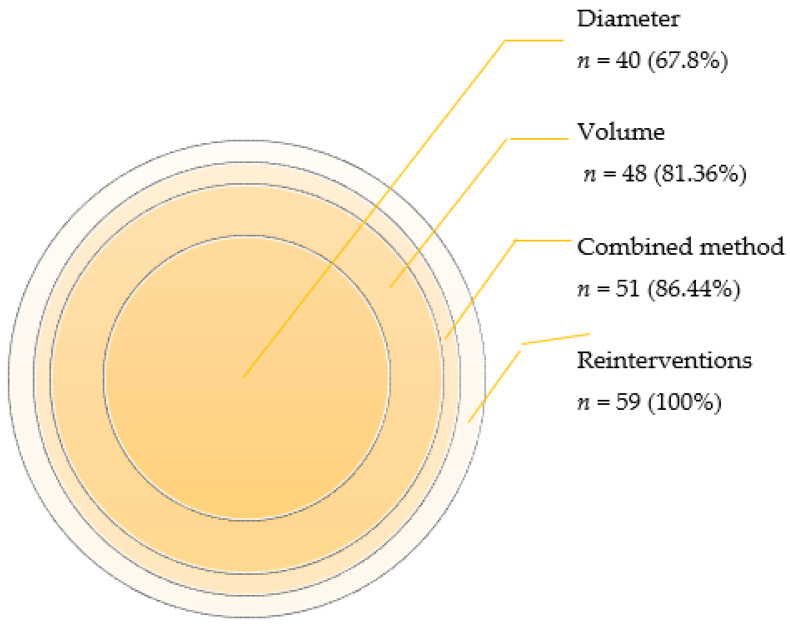
Number and percentage of cases with negative changes detected using each method.

**Table 1 jcm-12-06733-t001:** Co-morbidities to AAA.

Disease	Number of Patients	%
Hypertension	48	94.12
Diabetes Mellitus type 2	5	9.80
Asthma	1	1.96
Chronic Obstructive Airway Disease	2	3.92
Peripheral Artery Disease	10	19.61
Ischemic Heart Disease	29	56.86
TIA/Stroke	5	9.80
Chronic Kidney Disease	4	7.84
Atrial Fibrillation	4	7.84

**Table 2 jcm-12-06733-t002:** Remodeling of AAA in entire population.

Variable (*n* = 59)	Mean	Median	SD
AAA Diameter (mm)	69.222	68.000	17.980
Change in Diameter (mm)	5.008	5.770	14.328
AAA Volume (cm^3^)	292.545	252.764	171.746
Change in Volume (cm^3^)	152.566	109.532	196.376

AAA. Abdominal Aortic Aneurysm; SD. Standard Deviation.

**Table 3 jcm-12-06733-t003:** Changes in the sac of AAA according to changes in MTD.

	Regression of MTD		Increase in MTD		
	*n* = 19		*n* = 40		
Variable	Mean	SD	Mean	SD	*p* Value
AAA Volume (cm^3^)	192.996	113.134	339.830	175.608	<0.00 *
Change in volume (cm^3^)	49.713	168.553	201.421	191.407	<0.00 **
AAA Diameter (mm)	55.956	14.878	75.523	15.879	<0.00 *

AAA. Abdominal Aortic Aneurysm; MTD. Maximum Transverse Diameter; *. Mann–Whitney U test; **. Student’s *t*-distribution.

**Table 4 jcm-12-06733-t004:** Changes in diameter and volume based on type of endoleak.

	EL I N = 12			EL II N = 21			EL III N = 14			
Variable	Mean	Median	SD	Mean	Median	SD	Mean	Median	SD	*p* Value
Change in diameter (mm)	4.871	2.760	10.331	8.561	6.300	9.837	2.893	5.260	18.355	0.32
Change in volume (cm^3^)	114.454	97.122	114.694	145.864	129.587	155.933	171.945	141.179	248.860	0.83

EL I. Endoleak type 1; EL II. Endoleak type II; EL III. Endoleak type III; and SD. Standard Deviation.

## Data Availability

The data that support the findings of this study are available from the corresponding author upon reasonable request.

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
