# Peer review of "Beyond Diameter: Enhancing Abdominal Aortic Aneurysm Surveillance with Volumetric Assessments after Endovascular Aneurysm Repair (EVAR)"

_jcm, 2023, doi:10.3390/jcm12216733_

Round 1
Reviewer 1 Report (Previous Reviewer 1)
This is an interesting study looking at the utility of aneurysmal volumetric measurements in conjunction with maximum diameter measurements to assess aneurysm remodeling and endoleaks after EVAR. The authors previously submitted this manuscript and have made conceded efforts to better define the methodology and points of discussion around this work. However the study needs continued improvement across multiple areas, including significant revisions of the introduction, methods, and results prior to being considered for publication.
Introduction
Lines 40-42. This is a very odd place to define type II endoleaks. I would recommend moving this to the following paragraph. Additionally would recommend rephrasing as this sentence does not make sense as it currently reads “a type II endoleak occurs when … “
Lines 44-46. Would switch the wording of this. Define endoleak first then describe which is most common. Would also move the definition of type II endoleak here, where it fits much nicer (as stated previously).
Lines 62-71. Would change “complications” to endoleak, as this is a bit more accurate wording. I would recommend significant editing to this paragraph, first starting with removal of “we” as this is not very clear. Additionally, it is still unclear what “the current follow up strategy” is referring to. Is this your institution's strategy or are you taking this from SVS and/or European guidelines? I do see a citation to European guidelines but this needs to be more clear in the text. Of note, this is not in line with Society of Vascular Surgery guidelines. The statement “annual imaging after EVAR for all patients is not based on evidence and its not feasible, and high risk” is a pretty aggressive claim to make. Some evidence does exist to support this and in many institutions annual imaging is recommended and completed (thus making it feasible). Finally, I cannot agree that this is truly a “high-risk” practice. If the authors are going to make these claims I would want more citations supporting these claims than just the guidelines they cited.
Lines 72-76. I find this paragraph difficult to follow. When the authors state “there are a few indicators to measure risk of re-operation” this does not seem to fit the previous part of the sentence. What are the indicators? Or are the authors stating that there are “only a few” indicators to predict this? Lastly, what changes are you referring to in regards to volume?
Materials and Methods
No major edits suggested. The Methods are clearly presented.
Results
There is significant variation in the way the p value is reported. Recommend keeping these similar (p < 0.00 vs p < 0.001), given that p value was set at 0.05 I would favor the first. Please review p values and ensure these are consistently reported.
Throughout the entire results section there is vast inconsistency with number of decimal points. Decimal points that are reported should be calculated based on significant figures and should be consistent throughout tables and the text. Please correct.
Figures 2 and 3 seemed to be flipped in the text, the text for figure 2 appears to be referring to figure 3 and vice-versa. Please revise.
Line 191: this sentence appears to require editing, the word case is repeated.
Please remove text that discusses the results from the results section and only include this in the discussion. Line 197-198.
Lines 198-200. What is the p-value for this comparison?
Line 203-205. Where is the comparison? this is stating positive vs negative remodeling, what is the comparison between combined method and volume measurements.
Line 206. The text says Figure 3 but should read 4. Also the text reads sensitivity, but the figure does not show values of sensitivity rather percentage of total negative remodeling. These are different.
Line 213 and 221. Would re-phrase complications to “endoleaks”
Throughout the results and in points of the discussion the authors are inconsistent with how they state types of endoleaks. At points they state type II endoleak while at other they state Endoleak type II. I would recommend keeping these consistent. Please correct within figures as well.
Discussion
Line 304 change 83,3 to 83.3
AI as a helpful tool. I am having a hard time seeing how these studies connect to the present study. What does this have to do with the data you have presented?
Minor edits are required and are stated in above suggestions.
Author Response
Please see the attachment.

Reviewer 2 Report (Previous Reviewer 3)
references.
Line 58: Please add references.
Line 89: When you refer a material, you need to add not only the company but also the state and country. For example, for Medtronic is Santa Rosa, CA, USA.
Lines 92-93: They could be eliminated.
Please check once again the p values throughout the text.
Lines 191-195: This finding seems weird. The volume measurement was able to detect aneurysms that presented volume enlargement but decrease in MDT. This part needs clarification.
Line 228: This phrase seems incomplete. Please consider rephrasing.
Line 262: Positive changes is aneurysm decrease?
Discussion: It would be nice to decrease the length of the discussion and be a little more focused.
Just some parts needs further clarification to be completely understandable.
Author Response
Please see the attachment.

Reviewer 3 Report (Previous Reviewer 4)
Dear authors,
many thanks for resubmitting your modified research article entitled "Beyond Diameter: Enhancing Abdominal Aortic Aneurysm 2 Surveillance with Volumetric Assessments after Endovascular 3 Aneurysm Repair (EVAR)" in the Journal of Clinical Medicine.
Although the quality of writing and some remarks have been acknowledged, some major points remain that really do require your attention:
* In order to test the association of one model you have to have consecutively treated patients (in this case EVAR) plus those who experienced complications from the mentioned inclusion period, not only the one with complications. You cannot base your conclusions just based on the patients who experienced complications, this makes your assumptions biased.
* You mentioned that you did not have follow-up CTA for all patients since they were treated in another institution. This is a significant source of attrition/selection bias. How many patients were excluded due to this reason?
* What does it mean that you have 10 patients that control CTA for comparison with the baseline? Please explain, because it is hard to follow Figure 1.
Only minor points
Round 2
Reviewer 1 Report (Previous Reviewer 1)
Thank you for your submission and revision of this work. This is an interesting study looking at the utility of aneurysmal volumetric measurements in conjunction with maximum diameter measurements to assess aneurysm remodeling and endoleaks after EVAR.
Author Response
Dear Reviewer,
We would like to extend our sincere appreciation for your thoughtful and thorough review of our manuscript. Your valuable insights and constructive feedback have been instrumental in improving the quality and impact of our work.
Your rigorous review process has undoubtedly strengthened our work, and we are grateful for your guidance and support. We believe that the revised manuscript will make a meaningful contribution to the scientific community and advance the understanding of aneurysm assessment and management after EVAR.
Once again, we thank you for your dedication to the peer review process and for recognizing the potential of our research. We look forward to seeing our work published and contributing to the body of knowledge in this area.
Reviewer 3 Report (Previous Reviewer 4)
Dear authors,
many thanks for resubmitting your manuscript to the Journal of Clinical Evidence. Although the quality of the manuscript has improved a lot, still some points remain that require your attention. Major issues are labeled with asterix.
* The introduction and discussion are still quite long are require shortening and focusing the attention of your research article, please amend this.
- Lines 114-120 should go into the limitations, not the methodology
- Figure 1 should be made clearer, please include all numbers starting from 71 up to 51. Make it like a PRISMA flowchart for systematic review.
- Table 3, in the MTD regression group, do all parameters have a negative mean?
Only some minor points
Author Response
Dear Reviewer,
Thank you for your feedback. We appreciate your input and have addressed the issues raised. Here are our responses and the required revisions:
1. We have shorten introduction and discussion.
2. Those lines were moved to methodology.
3. Flowchart has been remade to resemble PRISMA flowchart.
4. In Table 3, the positions of numbers were inadvertently swapped between the "Regression of MTD" and "Increase of MTD" groups. This has now been corrected to accurately represent the data. All parameters have a positve number of mean.AAA volume and diameter depict mean values in AAA in which was regression of MTD. As for change in volume min. value was -255,136cm^3 and max 347,386cm^3. The mean value showed that although there was decrease in MTD, taking in consideration change in volume those AAA were enlarging in fact. That's also why volume should be added to MTD as a parameter to depict growth of AAA. After your suggestion to look in table 3 we also noticed that in table 2 there was wrong value of median, which was corrected.
We appreciate your thorough review and constructive feedback, which were immensely helpful in improving the quality and clarity of our manuscript. We promptly made these necessary revisions to address the major issues you identified and submitted the revised manuscript for your further consideration.
Thank you for your continued support and guidance throughout this process.
This manuscript is a resubmission of an earlier submission. The following is a list of the peer review reports and author responses from that submission.
Round 1
Reviewer 1 Report
This is an interesting study looking at the utility of aneurysmal volumetric measurements in conjunction with maximum diameter measurements to assess aneurysm remodeling and endoleaks after EVAR. The study needs improvement across multiple areas, including significant revisions of the introduction, methods, and results prior to being considered for publication.
Introduction
Page 1 Lines 31-33. This sentence needs editing. There are two separate statements here “EVAR is a leading treatment method for patients with AAA” and “EVAR has a short-term survival benefit when compared to open surgical repair.” I recommend re-phrasing these statements, as they currently read it is quite confusing.
Page 1 Lines 33-36. This is a run-on sentence. This needs significant editing and re-phrasing.
Page 1 Lines 31-38. I would recommend doing a larger search of the literature, as more recent studies have demonstrated similar long-term survival benefits between EVAR and OSR (Lederle et al 2019, DOI: 10.1056/NEJMoa1715955). Please re-phrase these statements and include contemporary studies, which are important to keep in mind in this context.
Page 1 Line 38. I am not sure what this statement adds. Current recommendations are regular surveillance for both open AAA repair and EVAR. Would rephrase or remove.
Page 1 Lines 39-40. How do the authors define “complications” in this context. Surveillance imaging after EVAR is to detect endoleak and/or stent graft migration/malfunction. Many readers may read “complications” as a post-operative event such as myocardial infarct, stroke, etc. In this context that could be confusing.
Page 2 Line 42. Would define endoleaks in the introduction.
Page 2 Lines 44-45. Needs to be re-phrased. “An additional value of CTA is it’s suitability for being a sole modality for follow-up.”
Page 2 Lines 47-52. Would recommend from using “we.” Recommend re-phrasing “In the absence of contrast, CT can be used to look for MTD enlargement and sac volume, which may be suggestive of an underling endoleak.”
Page 2 Line 51-52. Please define negative remodeling.
Page 2 Lines 52-53. This is not a sentence, please rephrase.
Page 2 Line 53-55. Whose current follow up strategy? Is this your institution's strategy or are you taking this from SVS and/or European guidelines? I would recommend citing one or both of these guidelines and referring to them here. Also would re-phrase to “the main goal of the first examination is to.” Again confused by the word “complication.”
Page 2 Lines 55-62. “Based on the first…” Again whose guidelines is this based off of? Is this your institutional protocol. Please clarify and re-phrase. Of note, this is not in line with Society of Vascular Surgery recommendations.
Page 2 Lines 65-66. Should read “The most common long-term complication of EVAR are endoleaks”
Page 2 Lines 67-68. Should be “from” not form
Page 2 Lines 68-69. Please re-phrase. Perhaps the authors meant “Type II occurs when the blood flows retrograde to the sac via a branch vessel.”
Page 2 Lines 70-71. Please break this sentence up, “Type IV endoleaks often do not require reintervention”
Page 2 Line 75. “Gold standard” and “AAA progression” please re-phrase second part of sentence as this does not make sense.
Materials and Methods
Page 3 Line 87-88. Why did the authors only choose to include patients who had re-operations from endoleaks? Why exclude patients who had endoleaks but did not have an operation?
Page 3 Lines 89-90. Please expand on inclusion and exclusion criteria. Were fenestrated/branched EVAR excluded, how about chimney EVAR, were emergent and elective cases included, how about ruptures?
Page 3 Lines 92-93. How can you ensure these phases would capture an endoleak? Typical endoleak imaging protocols would require delayed phase imaging.
Page 3 Lines 98-99. Please be consistent with abbreviations. You have already used CTA so please use “CTA” here.
Page 3 Lines 102-104. Can the authors expand on why they did not use a centerline measurement? That is the gold standard to obtain a maximum diameter measurement.
Page 3 Lines 113-114. Can the authors expand on why they chose to obtain a volume of the excluded aneurysm sac and the flow lumen? In an excluded aneurysm (i.e. that treated with EVAR) the volume of those two entities should ideally be separate (aneurysm sac is excluded) but in the presence of endoleak only the aneurysm sac would be expected to change in volume?
Page 3 Lines 115-116. Where was the proximal point of the segmentation?
Page 3 121-123. This needs to be defined earlier.
Results
If the p value was set at p < 0.05 then the authors should not be reporting p values with 4 significant digits.
Do the authors have information on type of endograft placed, ancillary procedures (renal stenting, endoanchors, proximal extension, re-ballooning), would also recommend providing information on index size at repair. All of these are important when considering endoleaks.
Please be consistent with use of “.” or “,” Please correct the tables to be consistent with what is stated in the text.
Throughout the entire results section there is vast inconsistency with number of decimal points. Decimal points that are reported should be calculated based on significant figures and should be consistent throughout tables and the text. Please correct.
Page 4 Line 135. This should read “Statistical analysis of patient population”
Please remove text that discusses the results from the results section and only include this in the discussion. Page 6 Lines 169-170. Page 7 Lines 187-189. Page 7 Line 200 and Page 8 Lines 201-202. Page 8 Lines 208-217.
Page 7 Line 174. Please be consistent with sac or sack. Please correct in table 3 as well.
Page 7 Line 182. Please add appropriate spacing “p-value < 0.0001”
Please re-phrase “endoleaks of type” to “type II endoleak” or “type III endoleak”
Discussion
Page 9 Lines 242-243. Screening tool for what?
Page 9 Lines 248-249. You cannot make this claim as you have not evaluated patient treatments or outcomes, please revise.
Page 9 Line 253. Please change have to has
Page 10 Line 271. Please change to AAA sac
Page 10 Lines 271-273. Would re-phrase, we do not ONLY repair aneurysms based on diameter, there are recommendations based on patient comorbidity burden and lifespan, additionally there are recommendations to consider repair of saccular aneurysms regardless of MTD.
Post-operative complications. How does your study add to the above study’s findings?
AI as a helpful tool. I am having a hard time seeing how these studies connect to the present study. What does this have to do with the data you have presented?
This manuscript requires significant editing of the english language. Please refer to the suggested edits that were provided.
Reviewer 2 Report
Dear authors,
Congratulations on your article “Beyond Diameter: Enhancing Abdominal Aortic Aneurysm Surveillance with Volumetric Assessments after Endovascular Aneurysm Repair (EVAR)”.
One of the strengths of this study is its objective and purpose. This is because it focuses on a relevant topic of the vascular surgery clinical practice, making a clear point of the practical application of its findings.
The introduction must be shortened and focused on the aims of the study, providing a straight perspective of what the overall study contains. It is also very important to contextualize the focus of the paper by summarizing information concerning the findings of other articles. Even though the objective is stated in the chapter, which is a positive point, it should be stated in a more clear way, since it is not explicit enough.
In the section of Materials and Methods some positive aspects include: Reference to the study design; Inclusion Criteria; Exclusion criteria. Nevertheless some less positive points are verified, as: No reference to the ethical commission approval; No reference to the blinding process. Beside this, the authors should state how the sample size was defined.
The Results chapter must be rewritten, since it is verified along all of it, interpretation of the results obtained. This section must indicate the results in an objective, impartial way, only by stating them. This is verified in the following areas: Page 6 line 164-172; Page 7 line 187-189; Page 8 line 208-217.
The authors should shorten the Discussion chapter and focus it on the results obtained in the previous section. Authors should also refer the strengths of this article when compared to what already was published so far. The paper does not adequately discuss and compare its findings with existing literature on AAA assessment and treatment. Without a comprehensive analysis of the current body of knowledge, it is difficult to determine the novelty and significance of the paper’s findings. The paper suggests that volume measurement may be a valuable parameter in AAA assessment and surveillance, but it does not clearly discuss the clinical implications of incorporating volume measurements into routine practice. How would the integration of volume measurement impact treatment decisions, and patient outcomes? This aspect needs to be addressed in more detail.
Best Regards,
Minor editing of English language required
Reviewer 3 Report
Lines 11-13: Do you the pre-operative values of MTD and volume were evaluated as perdictors of endoleak formation? Please clarify.
Lines 13-14: Do you mean that all cases needed re-intervention? Probably not. Please rephrase.
You evaluated the pre-operative scans in terms of diameter and volume to verify them as predictors of endoleaks. And then you made similar measurements in follow-up CT scans? Please try to clarify your methodology in abstract. I understand that there is a restriction in words used but it needs to be done.
Lines 15-16: Negative remodeling is sac expansion? Sac-expansion and stability? Please provide a definition or change wording at least in abstract. Which was your baseline method (MDT) and then volume analyses were compared. You need a referral method clearly stated.
Lines 17-18: “Volume measurement showed lower false negatives than MTD, indicating its effectiveness in 17 screening for negative remodeling.” Did you compare them statistically? If so, please provide a p value.
Lines 18-20: Combining both methods identified 51 negative remodeling cases and 8 positive changes, with higher sensitivity compared to MTD alone. 51 and 8…probably you are describing different time periods of follow-up. However, this phrase is confusing. Please consider rephrasing.
Introduction
Lines 30-31: Please add as reference the guidelines.
Line 36: “if the aneurysm is sealed properly, prevents enlargement of the aneurysm”. And type II endoleaks? Please consider rephrasing.
Lines 39-64: I am sorry but after the 1st month, guidelines suggest DUS as the main tool of surveillance. It is true that CTA is a great imaging modality to detect technical pitfalls but aneurysm diameter and presence of endoleak can be both detected by DUS too, especially in patients with renal failure. This paragraph is very large. Please consider rewriting. This paragraph should focus on the benefits of CTA during follow-up and classification of patients.
Lines 62-64: They do not really add something important. Please eliminate.
Lines 65-73: Could this part be incorporated to the first paragraph? In addition, providing definitions for the types of endoleak is not needed. Probably, if you want absolutely to report on that, add a figure.
Line 76: Please eliminate “increases”.
Line 81: why did you exclude type V endoleaks?
Line 80-82: Please make a bit more clear your aim. Relationship of MDT and volume pre- or post-operatively? Volume growth in referral to what exactly?
Line 86: Please rephrase to “between 2010 and December 31, 2021” and provide a specific date for 2010 as you did for 2021.
Lines 87-88: Did all patients undergo re-intervention? Please clarify if so. Why did you exclude patients with endoleak type II that did not need reintervention? Probably they were not transferred to your service, as you state later. It is quite of a bias… If all needed reintervention, meaning all had type I or III endoleak or type II with sac increase, then, in all cases sac expansion is suspected.
As EVAR you define the treatment using a standard commercially bifurcated device? Did you include chimney or fenestrated cases? Did you include patients with landing to the external iliac artery? Please clarify this information.
Please provide some information about the aneurysms. Were they only abdominal infrarenal? Did they extend to the iliacs? Howe many of them did so?
Line 93: What do you mean with vascular phase? Arterial? Delayed? Both?
Lines 93-95: Not needed to refer it. Please consider eliminating it. If you did the follow-up and had also the pre-operative scans, it does not matter if they were transferred or not.
Lines 117-118: “The segmentation endpoint was typically at the level of the common iliac artery or the aortic bifurcation, depending on the extent of the aneurysm.” It would not be an issue if landing was performed to iliacs and you evaluated volume down to iliacs, as also them tend to dilate during follow-up. This could increase the homogeneity of your cohort.
I am sorry but after reading the Methods, I still don’t understand your referral exam… It was the pre-operative scan, a post-operative scan. Which one? This was compared to what?
Another thing that is very important and it is not referred in methods has to do with the annotators? Who evaluated these scans? Was only one investigator? Multiple? Did you do an intra- or inter-observer variability analysis. Up to 5mm changes can be considered as “human” mistake. Your data should be evaluated by two investigators.
Please rename section 3.1. This “Patients’ cohort”.
Lines 145-146: Add them to Limitations section.
Lines 150-152 and 158-160: These lines seem more like a comment than a finding of the statistical analysis. Please add them in discussion.
Lines 157-158: Changes with what referral? The previous CTA…which was the pre-operative one? Or a post-operative? At 30-days? At what time-point of follow-up? Did you estimate the follow-up for the total cohort? Please add it in section 3.1.
Table 2: What set the indication to treat a 37mm aneurysm? You evaluated smaller aneurysms. Using which threshold? 50? 45mm? In minimum we see only one value, as well as in maximum. The difference is reported according to this one value. This table is not easy to understand.
Why variable is 59 in Table 2? Isn’t 51?
Lines 169-172: Please move these phrases to discussion.
Line 185: False negatives using what standard? I mean that having false negatives or positives, there is something standard to compare with. What was that?
Lines 192-197: What is the number of cases included? There is some type of inconsistency.
Lines 208-218: These are not findings but comments. Please move it to discussion.
Line 222: You state that you had sac decrease in type I endoleak?
For p values please add only two decimals.
Endoleak type I has been related both to sac increase and decrease. This finding incorporates a statistical error, as the finding seems irrational.
Lines 236-238: Throughout your results section, there are comments. Please eliminate them.
All references to be added at the end of the sentences.
Lines 309-331: This part could be shorter. Try to relate your findings to the literature. This part does not add a lot to your findings.
Lines 333-363: Interesting the future perspective but need to be shortened to a paragraph.
Minor revisions only.
Reviewer 4 Report
Dear authors,
many thanks for submitting your research article entitled "Beyond Diameter: Enhancing Abdominal Aortic Aneurysm 2 Surveillance with Volumetric Assessments after Endovascular 3 Aneurysm Repair (EVAR)" in the JCM. The authors introduce sac volume measurement as additional surrogate tool besides maximum aortic diameter for predicting late intervention due to type I-III endoleak. The topic is interesting. However, it contains some important points that require further attention from the authors. Major issues are labeled with asterix.
* In general, the introduction is quite lengthy. Please make it shorter.
- In line 52, this sentence does not make any sense. Please rephrase this.
- Line 55-61, you can cite the latest ESVS guidelines for the management of aortoiliac aneurysms-
* In the methodology, I miss one very important part dedicated to the explanation of your outcomes. This should be very clearly stated.
* Did you include all consecutively treated patients in your analysis? Can you present a study flowchart?
* Did all patients have infrarenal AAA? I miss one table describing important CTA AAA characteristics. Were all patients treated in an elective manner, or you had some urgent/emergent patients?
* I do not understand your methodology, did you use only 30-day CTA findings compared to the baseline CTA before the intervention in order to make assumptions about the positive/negative sac remodeling?
* Can you make one table where you would compare baseline and 30-day CTA?
* What was the duration of the follow-up? What is the protocol of follow-up in your institution?
* Please explain in the methodology what were these 3 groups.
- Line 145/146 - this should go in the discussion, i.e. limitations.
- In table 2/3/4, please report numerical data according to their normality distribution. Please avoid mentioning min, max, median, mean, etc. This makes the tables confusing.
* Table 2, you included 51 patients, now you have 59. Can you please explain this?
- In Table 3, can you please mention how many patients you have in patients who experienced an increase and decrease in MTD?
* Line 184/185 - What was the diagnostic gold standard measurement to detect sac shrinkage? To rephrase, how did you determine FN and specificity of volume/MTD measurement?
- Line 208-218 - this does not belong to the results section
* Discussion is very long and hard to follow. Please make it shorter. Sometimes less is more. E.g. AI part can be significantly reduced since this is not the topic of your paper.
* Limitations are very obscure. The majority of the text is describing the potential direction of future research, while the only minority is the real limitation of your manuscript. Please correct this.
No major, only minor
Round 2
Reviewer 1 Report
This is an interesting study looking at the utility of aneurysmal volumetric measurements in conjunction with maximum diameter measurements to assess aneurysm remodeling and endoleaks after EVAR. This research highlights important findings in a field of AAA research that is growing in importance. I believe this paper provides novel findings that would be of interest to this journals readers. However, prior to acceptance there are still minor edits that are required.
Page 2 Line 79 ïƒ delete extra “.”
Page 4 Line 128 ïƒ delete extra “.”
Table 1 ïƒ please ensure each % has two decimal points to keep this consistent
Table 2 ïƒ please ensure that decimal point “.” vs “,” are consistent also would recommend keeping number of decimal points consistent
Table 3 ïƒ Would recommend keeping number of decimal points consistent
Page 7 Line 183 ïƒ 67.80%
Page 7 Lines 184-186 ïƒ “false negative cases” and please change to 7.50%
Page 7 Line 187 ïƒ p-value of 0.00
Page 7 Lines 193-195 ïƒ 67.80% and p<0.00
Page 8 Lines 207-208, please keep number of decimal points consistent
Table 3 and 4 ïƒ please keep number of decimal points consistent
Page 9 Lines 242-244. This is very redundant. You just stated that volume was linked to MTD. How is this different than volume being associated with diameter changes. MTD is maximum transverse diameter. Would remove these sentences.
Reviewer 2 Report
Dear Authors,
I have nothing to add to your revised version.
Best Regards,
Joana Ferreira
Dear Authors,
Minor corrections of english are required.
Best Regards,
Joana Ferreira
Reviewer 3 Report
Dear Authors,
Thank you for letting me revise your work. I have a few suggestions.
I think that the aim of the study needs rephrasing. Probably: “This study aimed to investigate the relationship between maximum transverse diameter (MTD) and volume measurements, in patients who experienced complications after endovascular aneurysm repair (EVAR), and their association with the occurrence of endoleaks.” This would help to identify immediately the type of cohort and decrease the questions arising next into the text.
Another issue, that arises back is the fact that you report 51 cases and then the numbers do not conform (as your results are reported in the number of events rather than number of patients; as you stated in your responses to the reviewers, there were patients that underwent more than one reintervention). I think that this should be clarified already in abstract.
The introduction section is still very long. It needs to be shortened. For example, endoleak classification does not add a lot, considering that the reader has a baseline knowledge of vascular surgery.
Lines 32-33: Please add a reference (for example the guidelines).
Lines 40-41: This is not an absolutely correct statement, as it depicts the results of OVER trial but ignores any other RCT, than showed different findings. Please consider rephrasing.
Lines 48-55: Please add references. You cannot make statements, especially in the introduction without references.
Lines 48-49: As you state later, among the most significant purposes, is to detect sac enlargement during follow-up. This is a finding that drives decision making in many cases. Please consider rephrasing.
Lines 82-84: You study cohort is not really patients that underwent EVAR but rather patients who underwent re-intervention after EVAR for endoleak type I-III. I think that this should be clarified in text. The indication for re-operation was always some type of endoleak, and for type II an increased sac diameter? Am I right? Please state in text.
Line 95: Please correct primal with primary throughout the text.
Lines 111-117: Three annotators evaluated the scans; one experienced and two students. I think that both the intra- and inter-observer variability test (Bland Altman) should be provided for both diameter and volume measurements.
Lines 146-148: An information that provokes confusion is that you state the number of patients (51), while the truth is you analyzed events. This should be somehow stated in the methods and clarified in the results. In addition, when a patient underwent two reinterventions for example, the referral CTA for the second reintervention was the 30-day post-EVAR or the one after the first reintervention.
Lines 149-153: There is no need to duplicate Table 1 in text. Please eliminate these lines.
Line 163: When p=0.00 then it should be written as, p<.001. Please modified throughout the text.
Line 179: Provide in methods a definition for “negative remodeling”.
Lines 190-198: This is the part the becomes quite confusing for the reader, and this is why you need to explain in methods your way of case assessment.
Lines 204-210: Did any of these patients presented an endoleak in the 30-day scan?If yes, how many, and what type of endoleak? In addition, something that is missing, and probably would be more useful than the comorbidities, is the anatomic characteristics of the aneurysms. Maybe a Table would be very helpful.
Line 230: Please complete this phrase with “after EVAR”.
Lines 260-288: Please modify this part. The aim is not to report the findings of other studies and your opinion on them. You need to compare your findings with theirs.
Line 315-317: How volume can be used for designing or deciding on an endograft? All graft measurements relay on diameter. Please modify the text.
Subsection 4.4: What does this section add to your manuscript? You did not investigate neither thrombus nor patent vessels. Please consider revising.
Some minor mistakes.
Reviewer 4 Report
Dear authors,
many thanks for submitting your revised version of the manuscript. The overall quality has improved a lot, but some points still remain that require your attention. Major issues are labeled with asterix.
* When you make some changes in the manuscript, please use the track changes option. In this manner, it is much easier and faster to follow the changes that have been made in the manuscript.
* I still miss the study flowchart, please create one for the sake of clarity
* You have to clearly outline and explain your outcomes.
- How many patients had type I/III endoleak on intraoperative control angiography?
* You should exclude patients that underwent intervention, and just analyze patients undergoing first-time EVAR.
* Please shorten the introduction and discussion
* Diagnostic accuracy analysis (FN/TP and specificity/sensitivity) should be removed, because there is no gold standard test to be compared with.
no major comments